# Grape Phylloxera Genetic Structure Reveals Root–Leaf Migration within Commercial Vineyards

**DOI:** 10.3390/insects12080697

**Published:** 2021-08-03

**Authors:** Jurrian Wilmink, Michael Breuer, Astrid Forneck

**Affiliations:** 1Department of Biology, State Institute of Viticulture and Enology, Merzhauser Str. 119, 79100 Freiburg, Germany; michael.breuer@wbi.bwl.de; 2Department of Crop Sciences, Institute of Viticulture and Pomology, University of Natural Resources and Life Sciences Vienna, Konrad Lorenz Str. 24, A-3430 Tulln, Austria; astrid.forneck@boku.ac.at

**Keywords:** *Vitis vinifera*, *Daktulosphaira vitifoliae*, SSR, parthenogenetic reproduction, galling insect, genotype

## Abstract

**Simple Summary:**

In most wine regions around the world, commercial vineyards are planted with *Vitis vinifera* scions grafted on grape phylloxera-tolerating rootstocks. Root-feeding phylloxera populations still thrive on such rootstocks and occasionally leaf-feeding phylloxera populations are observed. The cause for these foliar infestations is thought to reside at the thickets of abandoned rootstock vines that grow on the risers of vineyard terraces and constitute a different habitat with large leaf-feeding populations. Besides, it is unclear if root and leaf populations within commercial vineyards are genetically connected, which may indicate a process of adaption that could lead to large foliar phylloxera populations and better-adapted phylloxera biotypes. To shed light on these issues, phylloxera root- and leaf-feeding larvae from commercial vineyards and larvae from nearby thickets were genetically compared, focusing on population structure and genetic association. Our study showed that foliar populations in commercial vineyards not only originate from leaf-feeding populations on nearby abandoned rootstock vines, but also from root populations within the vineyard. The results suggest that sexual recombination is rare in the study area and that direct root–leaf migration creates population bottlenecks based on founder effects or host plant adaption.

**Abstract:**

Depending on their life cycle, grape phylloxera (*Daktulosphaira vitifoliae* Fitch) leaf-feeding populations are initiated through asexually produced offspring or sexual recombination. The vine’s initial foliar larvae may originate from root-feeding phylloxera or wind-drifted foliar larvae from other habitats. Though some studies have reported phylloxera leaf-feeding in commercial vineyards, it is still unclear if they are genetically distinct from the population structure of these two sources. Using seven SSR-markers, this study analyzed the genetic structure of phylloxera populations in commercial vineyards with different natural infestation scenarios and that of single-plant insect systems that exclude infestation by wind-drifted larvae. We saw that during the vegetation period, phylloxera populations predominately go through their asexual life cycle to migrate from roots to leaves. We provided evidence that such migrations do not exclusively occur through wind-drifted foliar populations from rootstock vines in abandoned thickets, but that root populations within commercial vineyards also migrate to establish *V. vinifera* leaf populations. Whereas the former scenario generates foliar populations with high genotypic diversity, the latter produces population bottlenecks through founder effects or phylloxera biotype selection pressure. We finally compared these population structures with those of populations in their native habitat in North America, using four microsatellite markers.

## 1. Introduction

Grape phylloxera (*Daktulosphaira vitifoliae* Fitch) (Hemiptera: Phylloxeridae) has a complex life cycle that is intertwined with that of its obligate host plant *Vitis* L. Phylloxera larvae penetrate their stylet into the plant tissue, from where they start sucking parenchymal plant sap. Putative salivary constituents transform local plant tissue into cells that concentrate starch and amino acids [1,2]. After finding a suitable feeding site, first instar larvae become sedentary for the rest of their one- to two-month lifespan [3]. During this time, they molt four times until reaching adulthood, from which point they continuously reproduce asexually until death. This cycle repeats itself three to ten times per growing season, each time producing up to 360 eggs per adult [1]. In most wine regions, when the growing season ends, phylloxera larvae bridge the winter by hibernating on the vine’s roots [4]. In addition to this asexual life cycle, grape phylloxera is also able to reproduce sexually by developing into alate sexuparae, which give rise to nonfeeding male and female sexual forms. After mating, the female sexual forms each lay a single egg that overwinters on the grapevine bark and hatches in spring [4].

Apart from their mode of reproduction, phylloxera’s life cycle can also be divided based on the infested plant organ. Larvae can feed throughout the year on the vine’s roots, where they can feed on two types of galls: tuberosities (formed on lignified, older roots) and nodosities (formed on root tips of fine roots). Tuberosities may greatly damage the vine by affecting water and nutrient transport and allowing soil pathogens to move inside the plant vascular system [1]. Tuberosity susceptibility is therefore the main reason for the global grafting of European grapevines (*Vitis vinifera* L.) onto resistant rootstock hybrids (crossed from tuberosity-resistant *Vitis* species). Nodosity feeding, on the other hand, takes place on both the European grapevine and virtually all of the rootstock hybrids that are globally being used in commercial viticulture [5]. Nodosities normally do not cause severe damage to the grapevine [6,7]. During the growing season, phylloxera can additionally migrate from the vine’s roots to its leaves and, from there, create leaf galls. American *Vitis* species (e.g., *Vitis riparia* or *Vitis rupestris*), and rootstock hybrids thereof, on which phylloxera larvae perform such root–leaf migrations, are often more susceptible to leaf galls than *V. vinifera* are [8]. Though leaf galls are less harmful than tuberosities, a high leaf gall infestation before grapevine blooming is known to cause yield reductions on “Seyval” hybrids (Seyve-Villard 5276) [9]. To our knowledge, such deteriorations have not been published for commercial vineyards planted with grafted *V. vinifera* grapevines.

As *V. vinifera* leaves are less susceptible to leaf gall formation, leaf-feeding is generally absent in commercial vineyards [10]. In such cases, phylloxera’s life cycle is limited to year-round root gall feeding. *Vitis vinifera* leaf gall outbreaks normally only occur when vineyard practices are neglected, leading to the formation of phylloxera-susceptible shoots and leaves from rootstock hybrids within the vineyard, or when nearby thickets, occasionally present on the risers of terraced vineyards, inadvertently house abandoned rootstock hybrid vines that grow vigorously throughout the growing season. The infestation pressure created by the high number of leaf-feeding phylloxera in these thickets also enables leaf gall formation on nearby *V. vinifera* vines in commercial vineyards [11,12].

Over the years, leaf gall outbreaks have occasionally been documented to occur on *V. vinifera* leaves in commercial vineyards throughout the world [13]. It is, however, not always clear whether infestations originated from within the vineyard [13] and whether the infested grapevine in question was located in a commercial vineyard [14,15]. For other outbreaks, it is known that susceptible grape varieties are planted nearby [12,16,17], or that rootstock hybrid foliage is present due to neglected vineyard practices [18]. Early studies in the literature have suggested that leaf gall formation on *V. vinifera* in commercial vineyards is only possible through wind-drifted foliar larvae. In these scenarios, leaf-feeding populations were thought to be initiated after sexual reproduction (so-called fundatrix generation), against which *V. vinifera* was thought to be resistant [19]. When this first leaf-feeding generation was formed on American *Vitis* species that are vulnerable to leaf galls (e.g., the *Vitis* species often used in rootstock hybrids), they could subsequently migrate to infest *V. vinifera* leaves [20].

Contrary to such wind-drifted leaf infestation outbreaks, there are some commercial *V. vinifera* vineyards in the wine region Baden in South-West Germany that have been observed to ostensibly undergo direct root–leaf migration on *V. vinifera*. Here, the leaf-feeding phylloxera larvae are thought to directly originate from root-feeding phylloxera, in the absence of a transitional leaf-feeding population on rootstock hybrids. A recent study on the genetic structure of grape phylloxera that was also carried out in the wine region Baden, and in nearby wine regions in Switzerland, showed that commercial vineyards planted with the susceptible *V. vinifera* hybrid “Léon Millot” can undergo such direct root–leaf migrations [21]. The authors furthermore concluded that sexual reproduction rarely took place, which suggests a root–leaf migration by asexually produced larvae.

Since the identification of four SSR markers for phylloxera identification, by Corrie et al. [22], they are increasingly being used as markers in population genetics studies of grape phylloxera (being well suited for association studies and the genotyping of individuals) [23]. SSRs were hereby used to differentiate phylloxera populations based on hostplant species and to differentiate spatial and temporal sampling levels [13,22,24,25]. However, for the understanding of leaf infestation outbreaks throughout the growing season, an intensive sampling and holistic analysis of how local phylloxera habitats in and around a commercial vineyard genetically relate to each other is still missing.

Our aim was to identify the differences in phylloxera population structure, between scenarios whereby adjacent thickets of vigorous wild-growing rootstock hybrids form potential sources for *V. vinifera* leaf infestations, and scenarios without such landscape structures (Figure 1). We hypothesize that both scenarios primarily undergo asexual reproduction, but that the inherent population structure is different. We hypothesize that *V. vinifera* leaf populations in scenario 2 may originate from root populations within the vineyard, as well as from leaf populations from nearby thickets of wild growing rootstock hybrids. We furthermore aimed to differentiate the population structure and genetic variety of single-vine phylloxera populations at the start of the growing season, compared with the artificial vineyard habitat and phylloxera’s native habitat. We hypothesize that the genetic diversity lowers from native to artificial habitat, due to the rarity of sexual recombination, and from root to leaf populations, due to a population bottleneck to infest the less-susceptible leaves in commercial vineyards under vineyard management.

## 2. Materials and Methods

### 2.1. Phylloxera Sampling

The study was conducted in six commercial vineyards located throughout the wine region Baden, Germany (Table 1). In the first year, carried out in 2018, phylloxera of four of these vineyards were sampled randomly. The two vineyards near the villages of Britzingen and Pfaffenweiler were planted with grafted *V. vinifera* scions on rootstock hybrids (*Vitis berlandieri* × *Vitis riparia*). These two vineyards have annually recurring leaf gall infestations that perpetuate throughout the growing season, without the presence of nearby leaf-feeding phylloxera populations on the leaves of rootstock hybrids or other sources in the vicinity of the vineyard (scenario 3). The third vineyard, located close to the village Ihringen, is also planted with grafted *V. vinifera* scions on rootstock hybrids (scenario 2), and is located next to a large thicket that is covered with abandoned rootstock hybrids that are susceptible to phylloxera leaf-feeding (scenario 1). The commercial vineyard in Bahlingen (scenario 2) is planted with a fungus-resistant grape variety (grape varieties with more than 85% *V. vinifera* parentage that are accepted as *V. vinifera* varieties in European catalogs [26]). As with the vineyard in Ihringen, phylloxera was also present on the foliage of rootstock hybrids near the Bahlingen vineyard. The 2018 phylloxera sampling was carried out by randomly appointing grapevines to collect from at the end of the season (at the start of September). This was carried out to ensure that all annually migrating phylloxera were incorporated into the population structure. For these randomly chosen grapevines, samples were taken from five different root tips and five different leaves. This sampling was chosen to enable root–leaf population comparisons of individual vines and to create hierarchical clustering of phylloxera populations from vine to vineyard level.

The two vineyards sampled in 2019 are also located near the village Bahlingen and are planted with fungus-resistant scions grafted on rootstock hybrids. To better understand the initial leaf infestation in spring and the mode of reproduction with which the migration process takes place (sexual versus asexual), sampling was carried out at the beginning of July. To exclude the potential migration from external sources, three single-plant phylloxera-proof netting systems were installed per vineyard. These nettings were constructed in spring, prior to the annual foliar infestation outbreak. They consisted of a single phylloxera-proof net that was hung over the wires of the vine’s trellis system and at both sides dug 20 cm deep in the ground and sealed at the sides. The root and leaf sampling in single-plant quarantine systems enabled the genetic identification of bottlenecks from the root to the leaf populations, and a prediction on whether these larvae go through sexual recombination prior to leaf-feeding. Moreover, to quantify the extent of migration from external sources, this sampling was conducted on the roots and leaves of the quarantined plants and the leaves of nearby grapevines outside the netting system.

### 2.2. DNA Isolation and Genotyping

Adult leaf-feeding larvae were gathered by opening leaf galls and root-feeding larvae by digging up grapevine roots. Individuals were sampled with a moist brush and individually conserved in the vineyard in reaction tubes with 70% ethanol. The samples were genetically identified using microsatellite markers (SSRs). DNA was isolated according to Forneck et al. [21]. Briefly, after gradual dilution in water, the sample material was homogenized by cryogenic grinding and purified with Chelex 100 (Bio-Rad). Samples then received an SSR-primer and GoTaq G2 Colorless Master Mix (Promega). In total, seven SSR-primers were used to identify the samples: *Phy_III_55, Phy_III_30* and *Phy_III_36* [27], *Dvit6* [28], *DV4* and *DV8* [29], and *DVSSR4* [30]. These were chosen to be in accordance with the standardized set of markers for phylloxera research, described by Forneck et al. [29]. After thermal cycling, allele calling was conducted with capillary electrophoresis (ABI Prism), using fluorescence markers (6-FAM and HEX), as described by Tello et al. [31].

### 2.3. Population Genetic Analyses

The population’s genotypic richness was expressed by the amount of unique multi-locus genotypes (MLGs) and the number of samples of that population (*n*), calculated as (MLG − 1)/(*n* − 1) according to Dorken and Eckert [32]. The Phylloxera population deviation from Hardy–Weinberg equilibrium (HWE), as well as observed and expected heterozygosity, were calculated according to Weir and Cockerham using GENEPOP 4.7 [33]. These values were calculated for each locus and with a weighted average over all loci (i.e., F_IS_ multi-locus). In addition, F_ST_ values were calculated, measuring the amount of allele fixation in subpopulations. These values were measured in a pairwise manner, relatively comparing putative subpopulations, based on hostplant organ (i.e., comparing phylloxera root and leaf populations). Using FSTAT 2.9.4, these calculations were made according to Weir and Cockerham, because of its superiority under heterogeneous sample sizes [34]. The population fixation per vineyard was statistically calculated with AMOVA-tests (analysis of molecular variance) with the program Arlequin 3.1 [35]. The AMOVA-tests were carried out with 1000 nonparametric permutations, comparing the genetic structure of leaf- and root-feeding phylloxera populations in and around a vineyard, according to Excoffier et al. [36]. Furthermore, using 10,000 simulations and not assuming HWE, the number of overrepresented MLGs (compared with a panmictic population) was statistically calculated, using MLGsim 2.0 [37]. According to Tello et al. [31], repeated MLGs with *p*_sex_ values of *p* < 0.01 were considered truly clonal. In the tables of this paper’s results section, these truly clonal repeated MLGs are depicted next to the total amount of repetitive MLGs (i.e., the difference is the amount of recurring MLGs that could have been created with sexual reproduction).

For a comparison between the introduced European habitats of this study’s sampling (i.e., managed grafted vineyards and wild growth in abandoned thickets) and phylloxera’s native habitat, a study comparison was made. Recent genetics studies concluded that grape phylloxera was introduced at least twice from their native habitat to Europe [31,38]. These studies concluded that the native hosts of these phylloxera populations are *Vitis riparia* and *Vitis labrusca* from the north-east coast of North America. Using four shared SSR markers, the population genetic analysis was extended to leaf-feeding phylloxera samples that came from this region and hostplants from Lund et al. [39] (a selection of the sample material that was used by Tello et al. [31]).

## 3. Results

### 3.1. Population Structure of Vineyard Scenarios

In the conducted sampling, 350 MLGs were identified from a total of 603 complete samples. The tested vineyards housed 93 MLGs that were present in more than one sample. The late-season 2018 samples were best differentiated along the axis of habitat scenarios, though differences were also visible between host plant-feeding organs (leaves versus roots). In line with our hypotheses, the latter axis of differentiation was more pronounced in the early-season 2019 sampling with netting systems.

Most of the samples with recurring MLGs were found in Britzingen and Pfaffenweiler, the two isolated vineyards (scenario 3). The genotypic richness of these vineyards was about 0.5 for both root- and leaf-feeding phylloxera (Table 2). This was much lower than that of the two vineyards with adjacent thickets covered with rootstock hybrid (RH) foliage in Ihringen and Bahlingen (scenario 2), which had an average genotypic richness of 0.8 for both root- and leaf-feeding phylloxera samples. The genotypic richness of these leaf populations was higher than those of the leaf sampling in the corresponding thickets (scenario 1).

In the 2018 field experiments, the most abundant genotype (MLG 42) was found in 17 phylloxera root samples in Britzingen (Appendix A). The second most abundant (MLG 14) was found in ten different samples in Britzingen, consisting of leaf samples from different vines, as well as root samples of the exact same grapevines. The presence of a single MLG on both root and leaf populations of the same plant was also seen for MLGs 16 and 22 in Britzingen. MLG 41 in Britzingen and MLG 83 in Pfaffenweiler were also feeding on both RH roots and *V. vinifera* (Vin) leaves within the same vineyard, albeit sampled from different host plants. None of such leaf-root combinations were found in Ihringen and Bahlingen. In Ihringen, however, MLGs 109 and 130 were found on both RH leaves and Vin leaves. The same was observed in Bahlingen, where MLG 212 was found on RH and fungus-resistant (FR) leaves (Figure 2).

The *p*_sex_ analysis in Table 2 revealed that, based on the amount of repeated MLGs, most of the populations underwent no sexual recombination. The leaf and root populations in Britzingen, however, showed two MLGs that may be a result of sexual recombination. The omnipresence of asexual reproduction is also visible through to the negative F_IS_ values (i.e., the excess of heterozygotes) in all subpopulations with a significant deviation from Hardy-Weinberg equilibrium.

A similar pattern was visible for the pairwise fixation index comparison. The vineyards in Britzingen and Pfaffenweiler showed low fixation indices of 0.02 and 0.09, between their leaf- and root-feeding populations (Figure 2). The same figure shows that the AMOVA tests between leaf and root phylloxera populations in both vineyards were nonetheless significantly different from one another. For Ihringen, though the fixation index between RH root-feeding and VIN leaf-feeding in the vineyard was also small, it was three times higher than that of the pairwise comparison between Vin and RH leaf-feeding populations in the nearby thicket. The leaf-feeding FR population in Bahlingen had the lowest fixation index with its RH root-feeding population, which was 0.01. This value was four times lower than the comparison between FR and RH leaf-feeding populations. The AMOVA test *p*-value in Bahlingen was also the highest of the four vineyards; however, the subpopulations were still significantly distinct from one another. The overall F_ST_ value between the subpopulations of all four vineyards was 0.12 (i.e., 88% of the genetic variation was within the subpopulations).

An analysis of the individual SSR markers revealed that the loci consisted of an average of 7.1 alleles (PhyIII55: 8; PhyIII30: 5; PhyIII36: 8; DV8: 5; Dvit6: 7; DVSSR4: 8; DV4: 9). Considering unequal sample size, allelic richness showed similar results between all subpopulations. Subpopulations with a high number of different alleles also revealed more alleles that were unique for that subpopulation (Table 2). In Britzingen and Pfaffenweiler, the root-feeding populations possessed most alleles. For Ihringen and Bahlingen, this was true for the RH leaf-feeding populations.

Taken together, a clear difference was visible between the genetic diversity of scenario 3 phylloxera populations on the one hand and scenarios 1 and 2 phylloxera populations on the other. Phylloxera with the same MLG was found between root- and leaf-feeding populations in scenario 3, but not in scenario 2. The same MLG, however, was found between the leaf-feeding populations of scenarios 1 and 2.

### 3.2. Leaf Infestation Outbreaks

None of the MLGs in the 2019 netting experiment in Bahlingen coincided with those of the 2018 experiments, even though one of the 2019 plots was located next to the 2018 Bahlingen vineyard. In the early season single-plant sampling, single MLGs were much more abundant (Figure 3). The genotypic richness was, therefore, extremely low, especially for the leaf samples within the enclosures (on average 0.19). The sampling of root-feeding phylloxera, on the other hand, showed a high MLG diversity for the few plants that underwent the intensive sampling, having an average genotypic richness of 0.65.

The allelic richness in this experimental setting was highest for the samples of root-feeding phylloxera, with a small number of differences between leaf populations that were sampled from within or outside the enclosure. For these leaf-feeding populations, the observed heterozygosity was much higher than expected, leading to F_IS_ values of −0.35 and −0.62 for enclosed and outside populations, respectively, and −0.09 for the root-feeding populations. The subpopulations of Bahlingen E2 showed a low amount of genetic variation (F_ST_) among all three subpopulations (0.07, 0.07, and 0.08). The other location (Bahlingen E1) had a low population fixation between the enclosed leaves and these plants’ roots (0.07), with a high amount of fixation for outside leaves, compared with the roots (0.28) and enclosed leaves (0.33).

The low genetic diversity of the enclosed leaf population indicates that only few larvae start the annual leaf infestation and grow in number, creating a population bottleneck. The leaf-feeding population outside of the enclosure also showed a lower genetic diversity, although this was higher than that of phylloxera within the enclosure. The more negative F_IS_ values of the leaf populations are in line with these findings.

### 3.3. Reproduction and Genetic Diversity

A comparison between the single-plant samplings, vineyard samplings, and a sampling from phylloxera’s native habitat was conducted to obtain insight into the differences in genetic diversity and the principal mode of reproduction between three different levels of sampling, and to compare the population structure of the artificial habitats of European commercial vineyards with that of its native habitat in North America (Table 3).

Relative to one another, the three population analyses show distinct differences, whereby a gradual change can be seen, from single-plant to native habitat sampling. Due to the high amount of recurring MLGs, the genotypic diversity of the 2019 sampling was lowest, followed by that of the vineyard sampling. The reduction in analyzed SSRs from seven to four lowered the amount of significantly clonal repeated MLGs in this analysis compared to Table 2. Furthermore, even though the number of MLGs in the native habitat was a fifth of that of the 2018 sampling, the number of alleles was the same for both samplings. Conversely, the 2019 sampling, with a quarter of the MLGs of 2018, presented about half of its alleles. For eight different alleles, the lengths of the PhyIII55 and PhyIII30 markers were a single base pair longer in the 2018 and 2019 sampling than in the study of Lund et al. [39]. In contrast, the alleles of Dvit6 were more similar, covering five out of six shared alleles. Lastly, the inbreeding factor was positive and highest for the native sampling, around zero in the vineyard sampling, and highly negative for the single-plant sampling.

The higher genetic diversity and more positive fixation index indicate that the genetic diversity was much higher in phylloxera’s native habitat. The high clonal richness underlines this and may indicate a higher rate of sexual recombination. The single nucleotide difference between the alleles from this study and the study of Lund et al. [39] may be caused by a methodological difference in allele calling, or indicate a stepwise mutation.

## 4. Discussion

Using seven SSR markers, the late-season 2018 sampling was carried out to identify differences in phylloxera population structure across vineyards with different potential infestation sources. Our findings show that the annual population structure is affected by locally available sources of inoculum, whereby the infested grapevine species and the presence of abandoned thickets covered with wild growing rootstock hybrids play a central role for the infestation pattern. The early-season 2019 sampling with single-plant phylloxera exclusion netting systems revealed that root-to-leaf migration is predominately part of phylloxera’s asexual lifecycle and that only few MLGs start these annual foliar outbreaks. Furthermore, a comparison between the single-plant and vineyard sampling of this study, with data from a study that was conducted in phylloxera’s native habitat, allowed comparisons of sampling scale and between populations in native and introduced habitats. Here, we saw that the phylloxera populations in our study had a lower allelic richness and genetic diversity than the native population did.

### 4.1. Sources of Vineyard Leaf Infestation

The studied vineyards in Britzingen and Pfaffenweiler were planted with *V. vinifera* grafted on rootstock hybrids and had no external phylloxera infestation sources in their vicinity (scenario 3). Sampling in these vineyards was characterized by a high amount of repetitive MLGs for both root and leaf populations. In these vineyards, individual plants were often dominated by few MLGs. Some of these on both leaves and roots of the same vine confirm, for the first time, a direct asexual migration from rootstock hybrid root-feeding to *V. vinifera* leaf-feeding in commercial vineyards. In comparison, in Australian vineyards, where climate and quarantine measures severely reduce phylloxera migration, more fixation was seen between phylloxera root and leaf populations (based on four SSR markers) than in our study [40]. Another study in Australia also found MLGs that were adapted to root-feeding only [41]. In our study, MLG 42 in the studied vineyard in Britzingen showed this same limitation to root-feeding. Moreover, many MLGs were highly abundant belowground but not aboveground and vice versa. Though phylloxera larvae in this study were not restricted to migrate between plant-feeding tissue, population structure was visible based on different hostplant feeding niches (i.e., root- versus leaf-feeding). Phylloxera populations are in fact known to belong to biotypes with different hostplant adaptations [15]. Population structure may alternatively be the result of annual founder effects, which are known to heavily limit the genetic diversity that is present in a single growing season and location for asexually reproducing pests [42].

Though the genotypic diversity was low, the studied phylloxera populations revealed that the allelic richness was high. The exact same SSR markers that were used in this study were also used in other studies on phylloxera genetics in Europe. Comparing these studies, the mean number of 7.1 alleles per locus we found in 350 MLGs was higher than the 6 in 203 MLGs by Forneck et al. [21] and the 6.1 and 5.7 that Tello et al. [31] recorded for their European phylloxera groups of 143 and 159 MLGs, respectively. Considering the higher number of MLGs in this study (and with it, the chance to find rare alleles), the allelic richness is similar to those of these studies. Indeed, many loci showed some alleles that were only present in less than ten samples of our total dataset. For aphids that produce extremely high numbers of asexually reproduced individuals per growing season, it was shown that mutations occur frequently, leading to new alleles [43]. The high number of asexually reproduced offspring that is produced in a single season (especially due to leaf feeding) makes it highly likely that this is also applicable for grape phylloxera.

Comparing all seven markers, there were two MLGs (in Britzingen and Bahlingen) that were identical with two MLGs found in vineyards around Zürich (Switzerland) by Forneck et al. [21]. In both studies, these MLGs were, however, only observed in a single sample and are, therefore, not thought to resemble the characteristics of broad-spectrum super-clones [44]. Moreover, the most abundant MLGs in this study, MLGs 14 and 42, did not show the widespread presence of a general-purpose MLG, as described in the literature [1].

When the genetic variation was compared between subpopulations (based on F_ST_-values), 90 to 99.5% of all variation was shown to be present within the subpopulations, representing little fixation between subpopulations that feed on vineyard roots, vineyard leaves, and thicket leaves, in the same vineyard setting. These findings are in line with an inter-vineyard analysis conducted by Forneck et al. [21], where no significant population structure based on grape variety could be observed. A European-wide phylloxera sampling, which also included samples from Baden, found that (based on AFLPs) only two clusters could be identified for the whole continent [45]. Later studies, based on SSR markers and phylloxera’s genome, confirmed these finding [31,38]. Additionally in South America, in a study with 17 SSR markers, a lack of population structure was visible, with a genetic variation among Argentinean root-feeding populations of only 1% of total genetic variance, confirming our inter-vineyard results [25]. However, even though fixation is low, the AMOVA-tests retrieved significantly different leaf and root populations. In all, these findings seem to back the abovementioned argument for founder effects instead of biotype effects. It should, however, be noted that the genetic or biological background of phylloxera biotypes is still unknown and not represented by these SSR markers.

The pairwise F_ST_ comparison showed that the leaf and root subpopulations in all vineyards have a low rate of fixation compared with the total vineyard population. In Ihringen, the three-times-higher pairwise F_ST_ between leaves and roots, compared with *V. vinifera* and rootstock hybrid leaves, confirmed that the latter two are genetically more closely linked to each other. The foliar phylloxera populations of *V. vinifera* and rootstock hybrid leaves also housed phylloxera with the same MLG. Probably due to the higher susceptibility of the fungus-resistant variety that was cultivated in Bahlingen, the pairwise F_ST_ between leaf and root phylloxera was contrariwise lower than that between both leaf subpopulations. We thus see that successful colonization does not appear homogeneously in all four vineyard settings and seem to depend on the presence of phylloxera populations in and around the vineyard and the planted grape varieties. In a recent study on the transcriptome of two phylloxera lines with a host adaption to different *Vitis* spp., it was seen that each phylloxera line transcribed different olfactory genes in the search for a feeding spot [46]. Whether phylloxera larvae actively use this ability to migrate between the different grapevine species in a vineyard setting or if this occurs through random dispersal is still unknown (cf. [18]).

### 4.2. Population Bottlenecks

The early-season 2019 experiment with grapevine netting systems confirmed that founder effects are important for foliar phylloxera populations. In this experiment, the high amount of repetitive MLGs on the leaves from inside, as well as outside, the netting system at the beginning of the season show that few initial MLGs start the foliar outbreak and asexually grow in numbers. A multiple-year experiment is needed to confirm whether these MLGs are a selection of phylloxera larvae that are better able to initiate leaf gall outbreaks on *V. vinifera*, or if it is solely a founder effect and migrating MLGs are annually selected at random.

With these findings, the 2019 experiment also indicates that the root–leaf migration is predominately asexual. Relatively compared to its root population, the enclosed leaf population showed a lower genotypic diversity and a more negative inbreeding coefficient. Based on the 2018 *p*_sex_-values, sexual recombination should, however, not be completely ruled out and might occasionally occur. These findings are in line with other studies on phylloxera populations in Europe, which state that sexual recombination could rarely take place [28].

With the initiation of leaf infestations, phylloxera larvae of later generations were observed to heavily crowd that same shoot, without any sign of interspecific competition. Leaf-feeding phylloxera show increased gene expressions related to reproduction, compared to root-feeding phylloxera, which may facilitate this crowding effect [47]. Other studies on phylloxera leaf galling have shown that, by locally increasing shoot vigor, initial galling seemed to improve future galling on that same shoot by the formation of meristematic leaves [48]. This was also visualized in a ^14^C CO_2_ experiment, where the flow of photo-assimilates increased toward galled leaves [49]. Possibly due to this local compensation of nutrients, intraspecific competition plays a minor role for leaf-feeding populations [50]. This process may result in an increased reproductive ability of individual MLGs once they overcome the initial colonization barrier on *V. vinifera* leaves, augmenting the population bottleneck effect that is already present due to either one of the two aforementioned possible causes for population structure between root and leaf population. The netting experiment confirms this, with few different MLGs on leaves both within and outside the enclosure and a highly versatile root-feeding population feeding on single grapevine root systems.

Population bottlenecks due to root-to-leaf migrations do not explain the low genotypic diversity for the Britzingen and Pfaffenweiler root-feeding populations (scenario 3), compared with those in Ihringen and Bahlingen (scenario 2). In this regard, the thickets with rootstock hybrids that are present adjacent to the latter two vineyards (scenario 1) may be functioning as long-term genetic preservation for phylloxera populations. Besides the susceptibility of rootstock hybrid leaves, these were also not managed (e.g., hedged) throughout the growing season, contributing to high leaf-feeding populations. The vineyard vines themselves, on the other hand, were all younger than ten years, which could explain this lower genotypic diversity. Without human contribution, phylloxera is known to disperse only slowly throughout vineyards, migrating 15 to 27 m per year [51].

In Ihringen and Bahlingen, the root-to-leaf migration most likely took place indirectly, via a leaf infestation on rootstock hybrids (scenario 1). The vulnerable leaves of these hybrids likely did not create phylloxera selections for the initial leaf infestation, which resulted in the high amount of different MLGs that migrated from thicket to vineyard leaves. From the planting of these vineyard vines onward, these were already prone to a high diversity of phylloxera genotypes from the nearby thickets. The shared MLGs between leaf-feeding phylloxera in the thicket and vineyard and low F_ST_ values between the two populations confirm that the high diversity of MLGs can be ascribed to migrating phylloxera from the nearby thicket.

Field observations during the sampling of this study (data not shown) revealed that a high amount of foliar infestation correlated with a high amount of root infestation. On such plants, the highly abundant leaf MLGs did not correspond with those that were highly abundant on the roots (though the sampling was not exhaustive). Alternatively, vigorous plants are known to support higher populations of phylloxera, promoting both root- and leaf-feeding development [52], whereas phylloxera infestations can counteract this effect by lowering the total biomass of its host plant [53]. Besides, through plant signaling, leaf infestation may affect root infestation and vice versa [54]. To achieve this, leaf-feeding phylloxera are known to influence plant jasmonic acid expression in plant leaves [55]. These changes in foliar jasmonic acid in turn are known to impact the fertility of root-feeding phylloxera populations [56]. In the plant tissue of the root tips, root-feeding phylloxera themselves also actively suppress jasmonic acid levels while simultaneously increasing salicylic acid levels to preserve their feeding sites for further population increase [57]. It is therefore conceivable that leaf- and root-feeding populations may pose a crosstalk during hostplant interaction that promotes the two subpopulations.

### 4.3. Critical Considerations

Forneck et al. [21] and Tello et al. [31] (who used the same SSR markers as in this study) found F_IS_ values that were slightly positive for most populations. Conversely, we found here that all subpopulations had F_IS_ values that were slightly negative, i.e., more heterogeneous than expected (generally seen as sign for a high rate of clonal reproduction). Other studies that used some of the markers that were used in this study also found negative F_IS_ values for these markers [13,27,28].

Comparing the individual plant sampling, the vineyard sampling, and the native habitat sampling, a stepwise increase in F_IS_ values and genotypic diversity is visible. This may be explained by the patch-like distribution of clonal MLGs, visible at the lower hierarchical level of single plants, the inhomogeneous dispersal in vineyards, and the decreased visibility of such patches for large-scale sampling. The positive F_IS_ in the native sampling population furthermore suggests that inbreeding and, thus, sexual recombination are more widespread in phylloxera’s native habitat. This reduction in sexual recombination events after the introduction into new habitats was also observed for the oleander aphid (*Aphis nerii* Fonscolombe) [42]. The individual sampling areas for this aphid constituted of a single MLG that could change from year to year, revealing dominant founder effects that are also thought to have taken place in our study (though the genetic variation is much higher for phylloxera in its introduced habitat). It is known that phylloxera’s invasion can be traced back to multiple phylloxera migration events to Europe [31]. The combination of a high number of asexually reproduced offspring (reducing genetic drift) with occasional sexual recombination (increasing genotypic variability) and an unknown number of mutations (creating new alleles) may be the cause for the genetic variation and population structure that is visible for phylloxera in Europe. More research is needed to confirm these hypotheses and to better understand the long-term effects of population structures with annual bottlenecks that we described for the different phylloxera population scenarios in this study’s commercial vineyard settings.

The combination of hierarchical, spatial, and temporal sampling scales showed that a comparison of phylloxera population dynamic studies is prone to methodological biases. Populations bottlenecks that are visible in the early season were not visible in the late season, and the patch-like distribution of MLGs at a fine scale appeared as random distributions at a broader sampling scale. We furthermore elucidated the need to differentiate between root- and leaf-feeding populations and the limits of host plant species differentiation (visualized through the differences of scenarios 2 and 3). Comparing Table 2 and Figure 3 with Table 3 furthermore shows that the interpretation of F-statistics (and recurring MLGs) highly depends on the amount of SSR markers used. Using a standard set of markers, proposed by Forneck et al. [29], would improve the comparison of phylloxera population genetics studies and enable a comparison of MLGs.

## 5. Conclusions

This study revealed that in Baden/Germany, in a small vineyard-size sampling scale or even on that of a single vine, the genetic variety of coexisting phylloxera populations is high. The fine-scale sampling revealed that seemingly similar cases of foliar phylloxera outbreaks display different population structures, which are based on founder effects, feeding traits of local phylloxera populations, the vulnerability of *Vitis* species, as well as other sources of phylloxera in the vineyard surroundings. This study furthermore helps to understand the irregular pattern of phylloxera leaf infestation outbreaks in commercial vineyards and revealed the root–leaf migration of asexually produced larvae that can successfully infest *V. vinifera* leaves in commercial vineyards.

## Figures and Tables

**Figure 1 insects-12-00697-f001:**

Possible phylloxera populations scenarios in commercial vineyards on leaves (green) and roots (brown) of rootstock hybrids (RH), *V. vinifera* (Vin), and fungus-resistant varieties (FR). Dashed arrows represent potential annual migration pathways of phylloxera larvae.

**Figure 2 insects-12-00697-f002:**
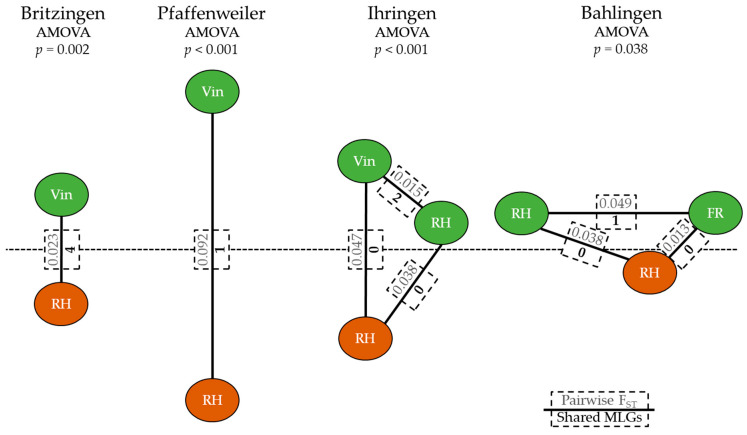
AMOVA *p*-values, pairwise F_ST_, and shared MLGs. Green and brown spheres represent phylloxera leaf and root populations sampled in 2018 from *V. vinifera* (Vin), fungus-resistant varieties (FR), and rootstock hybrids (RH). Lines indicate relative F_ST_ distances between the populations.

**Figure 3 insects-12-00697-f003:**
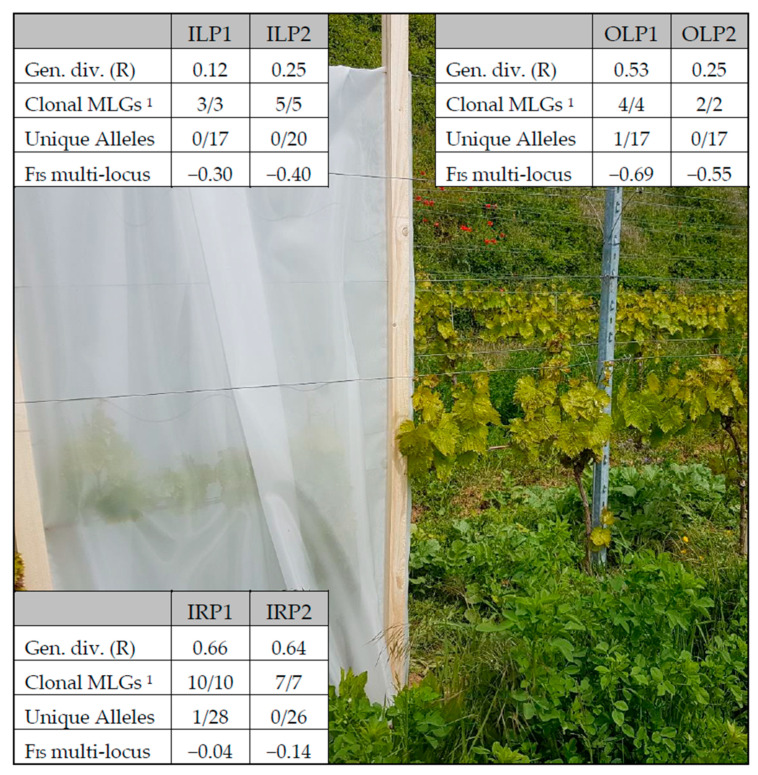
Genetic diversity parameters for phylloxera populations in the 2019 exclusion netting experiment. Samples were taken from leaf populations inside the enclosure (ILP), roots inside the enclosure (IRP), and leaves outside the enclosure (OLP), in the two vineyards Bahlingen E1 and E2. ^1^ Number of true clonal MLGs (*p*_sex_ < 0.01), per total recurring MLGs.

**Table 1 insects-12-00697-t001:** The amount of root- and leaf-feeding phylloxera larvae sampled from rootstock hybrids (RH), *V. vinifera* (Vin), and fungus-resistant varieties (FR), in different grafted commercial vineyards and surrounding thickets throughout the wine-producing region Baden, South-West Germany.

Sampling Type and Year	Vineyard Location	Vin + RH		FR + RH		RH (Thicket)		Total
		Leaf	Root	Leaf	Root	Leaf	Root	
Vineyard-wide sampling; 2018	Britzingen	100	85	-	-	-	-	185
	Pfaffenweiler	39	49	-	-	-	-	88
	Ihringen	71	65	-	-	45	-	181
	Bahlingen	9	8	85	13	25	-	140
Single-plant sampling; 2019	Bahlingen E1	-	-	50	55	-	-	105
	Bahlingen E2	-	-	60	63	-	-	123

**Table 2 insects-12-00697-t002:** Genetic diversity parameters for phylloxera leaf (L) and root (R) populations in vineyards and leaf populations in nearby thickets (T) covered with wild growing rootstocks, sampled in in 2018.

	Britzingen		Pfaffenweiler		Ihringen			Bahlingen		
	L	R	L	R	L	R	T	L	R	T
Individuals (*n*)	86	65	30	36	38	39	44	52	8	18
Distinct MLGs	39	36	16	19	36	28	37	46	8	12
Gen. div. (R)	0.45	0.55	0.52	0.51	0.95	0.71	0.84	0.88	1.00	0.65
Clonal MLGs ^1^	21/23	9/11	7/7	9/9	4/4	7/7	6/6	5/5	0/0	2/2
Unique alleles	0/25	5/33	0/23	0/25	0/28	1/35	3/37	0/32	0/20	0/27
*p* (HWE)	<0.001	<0.001	<0.001	<0.001	0.427	<0.001	0.005	0.17	0.587	<0.001
F_IS_ multi-locus	−0.05	−0.05	−0.13	−0.05	−0.05	−0.12	0.08	−0.04	0.05	−0.09

^1^ Number of true clonal MLGs (*p*_sex_ < 0.01), per total recurring MLGs.

**Table 3 insects-12-00697-t003:** Genetic diversity parameters based on four SSRs (PhyIII55, PhyIII30, PhyIII36, and Dvit6) for phylloxera populations in the 2018 and 2019 sampling (excluding samples outside the netting system), and for those sampled in their native habitat according to Tello et al. [31].

	Single-Plant Sampling	Vineyard Sampling	Native Habitat Sampling [39]
Individuals (*n*)	152	416	41
Distinct MLGs	45	186	38
Gen. div. (R)	0.29	0.45	0.93
Clonal MLGs ^1^	18/25	52/76	2/3
Unique alleles	0/17	8/28	22/28
*p* (HWE)	<0.001	<0.001	<0.001
F_IS_ multi-locus	−0.168	0.013	0.160

^1^ Number of true clonal MLGs (*p*_sex_ < 0.01), per total recurring MLGs.

## Data Availability

Data can be found within the article and the Appendix A.

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
