# Peer review of "Grape Phylloxera Genetic Structure Reveals Root–Leaf Migration within Commercial Vineyards"

_insects, 2021, doi:10.3390/insects12080697_

Round 1
Reviewer 1 Report
Dear Authors,
The study, the methods used, the description and conclusions and the bibliography used are excellent and the manuscript can be published in its current state (there are only a couple of typos). Congratulations.

Author Response
Thank you for your time and effort to review our manuscript. We are happy for the positive critical response and improved the typos that you highlighted.
Sincerely,
Jurrian Wilmink
Reviewer 2 Report
Authors of the present manuscript analyze seven SSR markers to study infestation route of grape phylloxera, Daktulosphaira vitifoliae. Interpreting statistical inferences, authors conclude that root-to-leaf migration of predominantly asexually reproducing grape phylloxera is a major route of leaf infestation.
Although most statistical analyses and interpretations were appropriate, the number of iterations run on MLGsim (lione 203) was limited to 1,000. I suggest increasing the number of simulations to at least 5,000 or 10,000. The inferences derived from a limited number of SSRs used may have somewhat affected the outcome to limit the value of the study. However, this manuscript is a good starting point until a better standard set of markers are developed.
Author Response
Thank you for your time and effort to review our manuscript. We are happy for the positive critical response and reran the MLGsim simulations with 10000 steps. Although differences were seen between the simulation runs, no significant differences at p<0.01 were seen. We are however grateful that the validity of our simulation output is now improved.
Sincerely,
Jurrian Wilmink
Reviewer 3 Report
line 44: Include taxonomic information of grape phylloxera
Line 78: Elaborate the genus name when starting the new sentence
Line 158: delete repeated "at the"
Line 212-215: this background information should go to introduction
Author Response
Thank you for your time and effort to review our manuscript. We revised our manuscript according to your critical points:
- We added the order and family name of grape phylloxera.
- We elaborated the genus name at the start of the sentence and checked the rest of the article for such cases.
- We removed the second "at the".
- We would however like to keep the specific information on the source plant and area of grape phylloxera in the materials and methods section, because it is directly related to the reason why we took this specific selection of MLGs from the Lund et al., research paper. The Tello et al., research paper also came to the conclusion of two migration events, based on a comparison of European grape phylloxera with the MLGs of Lund et al., in grape phylloxera's native habitat. We hope that, based on this information, you could place yourself in our perspective to keep these pieces of information together in the materials and methods section. They explain why we chose this method, and provide the citations to the used MLGs.
Sincerely,
Jurrian Wilmink
Round 2
Reviewer 2 Report
Authors of this manuscript has done a very good job presenting their data during the first round and have adequately addressed the points raised by reviewers. I recommend accepting the manuscript in the present form.
Reviewer 3 Report
Looks good to me